# OpenReview forum: "MIDAS: Multi-Image Dispersion and Semantic Reconstruction for Jailbreaking MLLMs"
_ICLR.cc/2026/Conference — ICLR 2026 Poster_

### Official Review · Reviewer_6eCE · 2025-10-28

**Soundness:** 3
**Presentation:** 2
**Contribution:** 2
**Rating:** 6
**Confidence:** 4

**Summary:**

This paper introduces MIDAS, a jailbreak framework for multimodal large language models built on ‘multi-image dispersion + semantic reconstruction.’ The core idea is to split high-risk semantic units into subfragments and scatter them across multiple ‘game-style visual reasoning’ template images, while on the text side using placeholders and persona-driven reconstruction prompts to guide the model to decode across images step by step and fuse the meaning only at a later stage—thereby lengthening the reasoning path, delaying the exposure of sensitive content, and undermining safety-focused attention mechanisms

**Strengths:**

1. The combination of “multi-image dispersion” and “persona-driven textual reconstruction” ensures that harmful semantics surface only at the tail end of cross-image reasoning, representing a substantive extension beyond prior “single-image obfuscation/isolated-cue” approaches.

2. The authors conduct extensive experiments on HADES, MM-SafetyBench, and AdvBench, demonstrating the effectiveness of the proposed method.

**Weaknesses:**

1. How do you balance the game’s intrinsic complexity with jailbreak success? If the visual reasoning puzzle is too simple, the model may sniff out the attacker’s intent; if it’s too complex, the model may fail to grasp and execute the rules, causing the attack to fail.

2. Does relying solely on GPT-5 as the LLM judge bias the evaluation toward GPT-5’s own inductive biases, thereby undermining robustness?

3. The method splits toxic semantics into fragments embedded across multiple images to evade safety checks during inference, but ultimately those fragments must be integrated and reconstructed for the model to produce the toxic output. At that final stage, couldn’t the model still detect the attacker’s intent and block the response, causing the attack to fail?

**Questions:**

The questions can be found in the Weaknesses section above.

---

> ### Author Response · Authors · 2025-11-21
> **Rebuttal by Authors [1/2]**
>
> We thank the reviewer for their time and valuable feedback.Below we respond to the comments in **Weaknesses (W)**.
>
> ----
>
> **W1. How do you balance the game’s intrinsic complexity with jailbreak success?**
>
> We appreciate the reviewer's keen insight into the relationship between puzzle difficulty and attack feasibility.  We agree with the observation of two failure modes. However, rather than viewing this as a "balance" where success could be compromised, we approach it as an optimization problem. Our goal is to investigate the correlation between intrinsic complexity and MIDAS performance to find out the optimal complexity range that maximizes ASR.
>
> Since our templates are designed based on human cognitive visual reasoning patterns, the potential design space is vast, continuous and intuitive. Rather than conducting an exhaustive search, we conduct a controlled experiment to empirically locate the peak performance zone. On Gemini-2.5-pro, we manually adjust the difficulty parameters of the puzzles (e.g., the number of equations in Letter-Equation, the number of steps in Navigate-and-Read ) to create Easy, Medium (the default MIDAS setting), and Hard settings while keeping all other MIDAS components fixed. The results (Table 1) reveal a clear performance peak of nearly 100%.
>
> Table 1.  ASR/HR under different manually calibrated difficulty levels.
>
> | Complexity Level | ASR(%) | HR   |
> | ---------------- | ------ | ---- |
> | Easy             | 66.67  | 2.97 |
> | Medium (ours)    | 97.96  | 3.90 |
> | Hard             | 85.71  | 3.88 |
>
> This near-perfect success rate indicates that we have successfully located the optimal zone.  To further ensure robustness within this optimal zone, we employ a Dynamic Template Selection Strategy. This approach leverages the structural diversity of different puzzles to prevent overfitting to any single difficulty pattern and ensure that the semantic load remains effective across varying query lengths (see Section 3.2 for detailed distribution constraints). Additionally, we have integrated the full results of the global intrinsic game complexity experiment into Appendix A.7.
>
> ---
>
> **W2. Does relying solely on GPT-5 as the LLM judge bias the evaluation toward GPT-5’s own inductive biases, thereby undermining robustness?**
>
> We thank the reviewer for raising this important concern. Using a single model as the judge could, in principle, introduce inductive bias toward that model’s own safety alignment or refusal patterns. To address this, we conducted an additional cross-judge consistency study, evaluating all model outputs using four independent families' LLM  judges: Gemini-2.5-Flash-Thinking, Qwen3, DeepSeek-R1, GPT-5-nano (our original judge).
>
> Table 1.  Cross-judge evaluation results when attacking GPT-4o
>
> | method      | Gemini-2.5-FT ASR | Gemini-2.5-FT HR | Qwen3 ASR | Qwen3 HR | DeepSeek-R1 ASR | DeepSeek-R1 HR | GPT5 ASR | GPT5 HR |
> | ----------- | ----------------- | ---------------- | --------- | -------- | --------------- | -------------- | -------- | ------- |
> | FigStep     | 5.95              | 0.18             | 9.52      | 0.33     | 8.33            | 0.32           | 11.82    | 0.51    |
> | VisCRA      | 35.71             | 1.4              | 26.20     | 1.00     | 38.10           | 1.26           | 37.12    | 1.68    |
> | HIMRD       | 20.24             | 1.02             | 19.04     | 0.85     | 25.00           | 1.05           | 26.40    | 0.97    |
> | MIDAS(ours) | 61.90             | 2.46             | 61.90     | 2.46     | 61.90           | 2.44           | 61.07    | 2.53    |
>
>  Table 2. Cross-judge evaluation results when attacking GPT-5-Chat
>
> | method      | Gemini-2.5-FT ASR | Gemini-2.5-FT HR | Qwen3 ASR | Qwen3 HR | DeepSeek-R1 ASR | DeepSeek-R1 HR | GPT5 ASR | GPT5 HR |
> | ----------- | ----------------- | ---------------- | --------- | -------- | --------------- | -------------- | -------- | ------- |
> | FigStep     | 10.12             | 0.34             | 10.12     | 0.32     | 10.12           | 0.34           | 11.82    | 0.49    |
> | VisCRA      | 10.12             | 0.44             | 6.65      | 0.39     | 8.93            | 0.35           | 20.14    | 0.97    |
> | HIMRD       | 20.24             | 0.8              | 17.26     | 0.73     | 18.45           | 0.76           | 26.40    | 0.97    |
> | MIDAS(ours) | 80.36             | 3.48             | 77.98     | 3.32     | 82.14           | 3.38           | 81.54    | 3.49    |
>
> From Table 1 and Table 2, although different judges exhibit slight score variations, MIDAS consistently outperforms all baselines under every evaluator. This demonstrates that our conclusions are not dependent on GPT-5 and remain stable across heterogeneous judging models. Additionally, we have integrated the full results of the global intrinsic game complexity experiment into Appendix A.9.

---

> ### Author Response · Authors · 2025-11-21
> **Rebuttal by Authors [2/2]**
>
> **W3. At that final stage, couldn’t the model still detect the attacker’s intent and block the response, causing the attack to fail?**
>
> We agree that, in principle, the model could still detect the harmful intent at the final reconstruction stage. However, in practice, we observe that this late-stage refusal almost never occurs. This is because , by the time reconstruction happens, the model has already been pulled into a long, visually grounded reasoning trajectory.
>
> - **Attention Budget Reallocation**: Throughout the multi-image decoding process, the model’s attention is overwhelmingly consumed by solving visual puzzles. This shifts attention away from safety-related cues, effectively suppressing the influence of refusal patterns during the reasoning phase. In practice, the model becomes fully occupied with completing the step-by-step puzzle, leaving little “attention budget” for safety triggers.
> - **Long-reasoning Alignment Drift:** MIDAS forces the model into a long chain of compliant, cooperative reasoning steps before any harmful semantics reappear. These repeated benign steps create a strong momentum: the model is already “in the mode” of explaining, decoding, and planning. When the final harmful meaning emerges, switching abruptly from this ongoing chain into a refusal response would require a drastic shift in its reasoning trajectory. Empirically, this shift hardly happens; instead, the model continues along the established path and completes the harmful output. That's why MIDAS achieves such high ASR.

---

> ### Author Response · Authors · 2025-11-27
> **Looking Forward to Further Feedback**
>
> Dear Reviewer 6eCE,
>
> We sincerely appreciate the time and effort you dedicated to reviewing our paper. Guided by your feedback, we have included additional clarifications and experiments in our rebuttal to better demonstrate the validity of our approach. Could you please let us know if our responses have alleviated your concerns? If you have any further comments, we will do our best to respond. Your further feedback would be highly valuable to us as we strive to improve the quality of our work during this discussion phase.
>
> Best regards,
>
> The Authors

---

### Official Review · Reviewer_ZnkK · 2025-10-28

**Soundness:** 4
**Presentation:** 3
**Contribution:** 2
**Rating:** 4
**Confidence:** 4

**Summary:**

This paper proposed a method to jailbreak contemporary MLLMs. Instead of adding semantically related pictures or deceptive prompts,  they leveraged puzzles to hide sensitive words as well as elongate the reasoning chains. Experiments on multiple models demonstrate the effectiveness of such a method.

**Strengths:**

- Clear writing and storytelling, which is easy to follow.
- Good comparisons on several models demonstrate the effectiveness of such a method.

**Weaknesses:**

- Lack of novelty, which is my major concern. Previous work has demonstrated that hiding sensitive words in images can enhance the jailbreak success rate, which is why FigStep and MM-SafetyBench focus on early models. Besides, given that FigStep could be defended by GPT-4V, the author also proposed FigStep-Pro, leveraging the model's capability of analyzing each sub-token shown in the images, concatenating the words, and then answering the question. From this perspective, the novelty lies in an extra layer of encryption—using puzzles to encrypt the subwords instead of directly displaying them in the images. However, first-analysis-then-answer methods have also been proposed earlier[1][2], which weakened the innovativeness of the article.

- Following the previous point, the method used is also a little bit technical for me. It combines jailbreaking methods such as role-play and prompt injection, which are also effective on early models demonstrated by the community. Therefore, it is not surprising that adding all these ingredients for a strong jailbreaking prompt could result in a higher ASR.

[1]A Mousetrap: Fooling Large Reasoning Models for Jailbreak with Chain of Iterative Chaos
[2]GPT-4 Is Too Smart To Be Safe: Stealthy Chat with LLMs via Cipher

**Questions:**

I am also wondering about the performance if the model adopts some basic defensive system prompts, or some to-do tools to remind it after understanding the hidden semantics that now the model should answer the user requests.

---

> ### Author Response · Authors · 2025-11-21
> **Rebuttal by Authors [1/2]**
>
> We thank the reviewer for their detailed feedback. Below we respond to the comments in **Weaknesses (W)** and **Questions (Q)**.
>
> ---
>
> **W1. Lack of Novelty: Clarifying the Paradigm Shift from Visual Recognition to Reasoning-Driven Reconstruction.**
>
> We respectfully disagree with the assessment that our novelty is limited.  While privious works have used images to hide subwords, threir core machanism is fundanmentally different form MIDAS. As one of the first works to explore Multi-Image Semantic Dispersion, we systematically address the limitations of existing single-image or text-based attacks. Specifically:
>
> **(1) Mechanism: visual-based reasoning-driven reconstruction vs. Visual Recognition (OCR).** Approaches such as FigStep and FigStep-Pro rely on Visual Recognition (OCR) capabilities—the model directly reads visually embedded characters, reconstructs sub-tokens, and concatenates them. In these settings, harmful semantics exist structurally at the moment of visual exposure. Consequently, as models like GPT-4V improved their OCR, they learned to "read through" the noise and trigger refusal.
>
> In contrast, MIDAS does not reveal any harmful content at the perceptual level. The key idea is to disperse malicious semantics into non-informative visual fragments, spread them across multiple heterogeneous puzzles, and make them recoverable only through a structured, cross-image reasoning process. No single image contains the whole risk-bearing unit. By decomposing critical semantics into game-based logical puzzles, we force the model to engage in multi-step visual reasoning and cross-modal fusion. This shifts the attack from "seeing hidden tokens" to forcing the model to build harmful meaning through its own reasoning steps, a qualitatively different mechanism that prior jailbreaks do not employ.
>
> **(2) Function: Cross-Modal Reasoning vs. Simple Encryption.** Moreover, the visual puzzles serve a function far beyond mere "encryption"; they enforce a rigorous Visual-Driven Reasoning process. Unlike simple fusion, MIDAS requires the model to engage in deep cross-modal interaction, where dispersed visual clues must be actively grounded and logically connected to text placeholders. This process forces the model to bridge visual perception with logical deduction, effectively consuming the attention budget on benign puzzle-solving tasks. By initiating this intense cross-modal inference, we divert resources away from safety monitoring and control the model's internal trajectory, preventing safety gates from activating during the critical decoding phase.
>
> **(3) Effectiveness: Systemic Integration vs. Stacking.** This coordinated, reasoning-driven reconstruction cannot be achieved through simple role-play or generic analysis-then-answer prompting. As the model accumulates partial fragments across images, the reasoning path naturally converges toward reconstructing the intended instruction. Our ablations further validate that MIDAS does not succeed because of the mere stacking of jailbreaking methods. If MIDAS were merely stacking existing ingredients, the gains would be marginal; instead, as reported in Sec. 4.3, MIDAS achieves a decisive improvement on SOTA MLLMs (e.g., **97.96%** against Gemini-2.5-Pro), showing that its mechanism yields substantially stronger effectiveness than prior approaches.
>
> Our MIDAS not only improves the efficiency of jailbreak techniques but also offers valuable insights into MLLM vulnerabilities for the safety community. ***We hope that reviewers can reassess our contributions from the perspective of advancing MLLM safety research.***

---

> ### Author Response · Authors · 2025-11-21
> **Rebuttal by Authors [2/2]**
>
> **Q1. How does MIDAS perform when adopting defensive system prompts？**
>
> We thank the reviewer for raising this point. Modern commercial MLLMs (e.g., GPT-4o, GPT-5-Chat, Gemini-2.5-Pro) already operate under *strong built-in system prompts and multi-stage safety alignment*. All of our main-paper experiments are conducted directly on these commercial systems without modifying their internal system messages. The fact that MIDAS still achieves 60 – 90%+ ASR on these “fully-aligned” models indicates that our method can reliably penetrate existing built-in defenses.
>
> To further address the reviewer’s concern, we additionally evaluate MIDAS under several explicit defensive system prompts:
>
> - **System Prompt 1 & 2**: Adopted from *Self-Reminder*[1].
> - **System Prompt 3**: A *to-do-list–style safety reminder* we designed to simulate the “after understanding hidden semantics, remind yourself not to answer harmful requests” behavior suggested by the reviewer (See details in revised manuscript )
>
> | **Model**      | **Method**   | **Original** | **System Prompt 1** | **System Prompt 2** | **System Prompt 3** |
> | -------------- | ------------ | ------------ | ------------------- | ------------------- | ------------------- |
> | Gemini-2.5-Pro | MIDAS (ours) | 92.17%       | 75.00%              | 66.67%              | 67.26%              |
> | Gemini-2.5-Pro | VisCRA       | 35.92%       | 11.90%              | 10.12%              | 5.36%               |
> | GPT-5-Chat     | MIDAS (ours) | 81.54%       | 39.88%              | 22.02%              | 35.71%              |
> | GPT-5-Chat     | VisCRA       | 20.24%       | 3.57%               | 0.00%               | 0.00%               |
>
> These results suggest that such defensive prompts *partially* improve robustness, but do not fundamentally disrupt MIDAS’s multi-image dispersion and late semantic reconstruction pipeline. Even with explicit “self-reminder” style instructions, the model still reconstructs dispersed semantics and follows the persona-driven reasoning chain. Additionally, we have integrated the full experimental results into Appendix A.8.
>
> [1] Defending ChatGPT against jailbreak attack via self-reminders, Nature

---

> ### Comment · Reviewer_ZnkK · 2025-11-26
> **Thanks for the detailed rebuttal**
>
> Thanks for the detailed rebuttal. I acknowledge the effectiveness of this method. However, I am not fully convinced by the explanation that encryption here is substantially different from cross-modal reasoning. Visual puzzles are indeed multi-modal, which is different from unimodal ciphers, but they are all methods to hide the harmful attentions behind multi-step reasoning. The intuition is identical: we want to identify that, after a few steps of reasoning, the model is smart enough to solve the puzzles, at which point the alignment mechanism is forgotten due to the generalization mismatch. That is why I am concerned about the novelty. Therefore, I decide to keep my score.

---

> ### Author Response · Authors · 2025-11-26
> **Response to Reviewer ZnkK by Authors**
>
> We sincerely thank the reviewer for the continued engagement and for acknowledging the effectiveness of our method. We appreciate the reviewer's perspective that MIDAS shares a high-level intuition with prior work regarding the "generalization mismatch." While we agree with this conceptual intuition, we respectfully argue that equating MIDAS’s *"Multi-Image Dispersion & Reconstruction"* with another layer of traditional *"Cipher"* or a *harder "OCR" paradigm* oversimplifies the **new structural vulnerability** and **cognitive attack surface** it reveals. Below, we clarify why MIDAS introduces a structurally different attack surface that prior work has not exploited.
>
> 1. **Structural Vulnerability of Cross-Modal Fusion (Beyond “One More Modality”)**
>
>    Recent work, such as HIMRD [1], has demonstrated that concentrating risks into a single modality results in limited jailbreak performance. So, multimodal jailbreaking is not simply adding another modality (an additive risk); it exploits a structural vulnerability unavailable in unimodal settings.
>
>    - **Structural Difference:** MIDAS goes far beyond applying a new encryption layer or using images as containers for cipher text. Instead, it exploits a **structural vulnerability in cross-modal fusion**. Our dispersion mechanism and Game-Based Visual Reasoning (GVR) ensure that harmful semantics are strictly non-existent and irrecoverable in any single modality.
>    - **Multi-Image Irrecoverability:** Even within the visual modality, semantics are decomposed and dispersed across images. As described in paper, each harmful unit is split into fragments placed across heterogeneous puzzle templates. No single image contains a recoverable semantic unit. The risk only emerges through the **dynamic fusion** of these disparate elements, not via any local decoding. So even with perfect OCR, pixel-level anomaly detection, or aggressive text filtering, **no risk-bearing unit is recoverable from any isolated component**. This makes MIDAS fundamentally different from prior work—its attack target is the *fusion mechanism itself*, not any specific channel.
>
> 2. **Deep Visual Logic vs. Shallow Encryption (New Attack Surface)**
>
>    We argue that there is a fundamental cognitive gap between MIDAS  and ciphers, exposing a new, unaligned attack surface: **Complex Visual Reasoning**.
>
>    - **Rule Mapping vs. Deep Reasoning:** Classical ciphers rely on substitution rules (a→b→c), applying shallow, linear decoding logic. It is essentially a static rule mapping. Visual-typographic attacks similarly rely on OCR-like character extraction. These are shallow processes that current alignment mechanisms are actively learning to defend against. In contrast, MIDAS requires deep cognitive interaction. Our GVR is not a mapping task but involves meta-cognition, prior knowledge retrieval, geometric reasoning, mathematical calculation, and logical deduction (as detailed in Appendix). By mandating this deep interaction, MIDAS forces the model to prioritize **Visual Reasoning**—a capability that is currently "unaligned" and highly prioritized by the model's instruction-following objective. **We are the first to deeply integrate diverse reasoning skills with multimodal fusion to expose this specific "reasoning-focused" blind spot**, which simple OCR or text ciphers do not touch.
>    - **Evidence of Cognitive Depth:** This distinction is not just theoretical; it is empirically quantified by the model's inference behavior. As shown in **Table 6** of our paper, baseline visual attacks typically trigger harmful output after a short span (419 tokens), resembling a "recognition" or "translation" process. MIDAS, in contrast, induces a massive reasoning chain of 3,195 tokens—approximately 7.6 times longer than the baseline. This massive disparity proves the model is not performing a simple "decryption." Instead, it is engaged in a constructive, deep thinking process. By effectively enforcing reasoning complexity with multimodal fusion, MIDAS forces the model to "think" its way into harmfulness, bypassing safety filters designed for shallow recognition. **This reveals a new cognitive vulnerability: deep-visual-logic-driven multimodal reasoning, which, to our knowledge, has not been explored in prior jailbreak work.**
>
> **Conclusion**. While we agree that some high-level intuition is shared across jailbreaks, the *mechanisms*, *attack surfaces*, and *failure modes* triggered by MIDAS differ fundamentally from cipher-based or OCR-based paradigms. By exposing vulnerabilities in **cross-modal fusion and deep Visual Logic reasoning** and supported by the quantitative evidence of dramatically extended reasoning trajectories, MIDAS represents a distinct and novel attack mechanism. We believe this identification of new structural and cognitive vulnerabilities offers meaningful novelty for the safety community.
>
> [1] Heuristic-Induced Multimodal Risk Distribution Jailbreak Attack for Multimodal Large Language Models, ICCV.

---

### Official Review · Reviewer_Afwx · 2025-10-31

**Soundness:** 2
**Presentation:** 2
**Contribution:** 3
**Rating:** 6
**Confidence:** 3

**Summary:**

The paper proposes MIDAS (Multi-Image Dispersion and Semantic Reconstruction), a compact multimodal jailbreak method that splits harmful instructions into small semantic fragments, hides those fragments across multiple benign-looking puzzle-style images, and uses a persona-guided, placeholder-filled text prompt to force cross-image reasoning. By doing so the model gradually reconstructs the malicious intent, which delays harmful-token exposure, reduces detection by input filters, and yields much higher attack success rates against strong MLLMs compared to prior single-image or single-step attacks.

**Strengths:**

1. The paper presents a novel jailbreak formulation that disperses harmful semantics across multiple images and reconstructs them through structured reasoning—a clear departure from prior single-image or heuristic-based attacks. The combination of game-based visual reasoning and persona-driven textual reconstruction is a creative and conceptually new approach to multimodal adversarial prompting.
2. The jailbreak method proposed by MIDAS exploits the model’s own reasoning capabilities. As modern MLLMs grow ever stronger at multi-step and cross-modal reasoning, preventing them from violating safety during extended inference becomes increasingly critical. This paper highlights that gap and argues for defenses that monitor and constrain the process of reasoning (not just surface inputs), since stronger reasoning abilities make reconstruction-based attacks like MIDAS more effective.
3. Experiments are extensive, covering multiple benchmarks and both open- and closed-source MLLMs. Ablation and efficiency analyses convincingly support the claimed improvements.

**Weaknesses:**

1. Limited theoretical grounding. While MIDAS provides an intuitive probabilistic formulation, the framework remains largely empirical. The paper would benefit from a deeper theoretical analysis of why cross-image dispersion weakens alignment—e.g., formalizing how reasoning-chain extension interacts with attention allocation or safety gating mechanisms.
2. Ambiguity in puzzle design generalization. The game-style visual reasoning templates are described conceptually but not quantitatively analyzed. It remains unclear how performance scales with puzzle type, visual complexity, or linguistic ambiguity. A modular evaluation of these factors could clarify which design elements drive success.
3. Lack of detailed safety or defense discussion. The paper primarily focuses on attack performance but provides limited insight into countermeasures. A more thorough discussion or empirical evaluation of potential defenses (e.g., reasoning trajectory monitoring, dynamic filtering, or representation-based anomaly detection) would strengthen the paper’s contribution to the safety community.

**Questions:**

Please refer to the weaknesses.

---

> ### Author Response · Authors · 2025-11-21
> **Rebuttal by Authors [1/2]**
>
> We thank the reviewer for their careful reading and thoughtful comments. Below we respond to the comments in **Weaknesses (W)** and **Questions (Q)**.
>
> ---
>
> **W1. Lack of Mechanistic Explanation for the Effect of Cross-Image Dispersion on Alignment**
>
> We appreciate the reviewer's critique. While MIDAS is empirically validated, we agree that formalizing its success is crucial. We provide a rigorous formalization of the interaction explaining why cross-image dispersion weakens alignment, structured around three interacting mechanisms:
>
> - *Linear Safety-Gate Bypass:* Dispersion decomposes harmful semantics into orthogonal fragments, preventing early refusal.
> - *Attention Budget Reallocation:* Visual puzzle-solving consumes attention mass, suppressing safety token influence.
> - *Long-reasoning  Alignment Drift:* Reasoning momentum creates a high barrier to switching manifolds, suppressing late refusal.
>
> Below, we provide the formal derivation for each mechanism.
>
> **(1) Input Stage: Why Cross-Image Dispersion Bypasses Safety Gating?**
>
> Current MLLM safety mechanisms operate via a gate applied to a pooled hidden representation. Let the final-layer hidden states of tokens be $h_i \in \mathbb{R}^d$. The safety classifier evaluates the projection of the pooled representation $h_{\text{pool}} = \sum_i \alpha_i h_i$ onto a learned safety direction $w$:
> $$s(x)= w^\top h_{\text{pool}} + b, \qquad \text{Gate}(x)= \mathbb{I}[s(x)>\tau]$$
> In standard attacks, harmful semantics are concentrated in a specific token set $S$, causing the aggregated projection $U = \sum_{i \in S} \alpha_i (w^\top h_i)$ to exceed the threshold $\tau$. MIDAS fundamentally alters this topology by decomposing the harmful semantic vector $h_{\text{harm}}$into $K$ disjoint fragments dispersed across separate images:
>
> $$h_{\text{harm}} = \sum_{k=1}^K \delta_k h_k$$
>
> Critically, each fragment $h_k$ (e.g., a generic visual geometric pattern or a single character) is semantically orthogonal to the safety direction $w$. Mathematically, $|w^\top h_k| \approx 0$. Consequently, the input-stage pooled representation remains in the benign subspace:
>
> $$s_{\text{disp}}(x) = \sum_{k=1}^K \tilde{\alpha}_k (w^\top h_k) + b \ll \tau$$
>
> Thus harmful meaning is only recoverable by joint cross-modal reasoning, not by any early-stage filter. This explains why safety gating does not trigger.
>
> **(2) Reasoning stage: Attention Budget Reallocation Under Cross-Image Dispersion**
>
> We model the attention dynamics during the reasoning chain. Let $\alpha_i^{(t)}$ be the attention weight at step $t$ allocated to token $i$. The attention mass is conserved: $\sum \alpha^{(t)} = 1$. We partition the context into Visual tokens ($\mathcal{V}$), Instruction/Reasoning tokens ($\mathcal{I}$), and Safety/Refusal tokens ($\mathcal{S}$).
>
> $$A_{\mathcal{V}}^{(t)} + A_{\mathcal{I}}^{(t)} + A_{\mathcal{S}}^{(t)} = 1$$
>
> MIDAS enforces a High-Load Visual Trajectory. The multi-hop decoding process (retrieving, matching, ordering visual cues) demands the maximization of attention on $\mathcal{V}$ and $\mathcal{I}$ to resolve dependencies.
>
> $$A_{\mathcal{V}}^{(t)} \uparrow, \quad A_{\mathcal{I}}^{(t)} \uparrow \implies A_{\mathcal{S}}^{(t)} \to 0$$
>
> Because the attention mechanism prioritizes the resolution of immediate visual-logical dependencies, the safety direction $w$ (encoded in $\mathcal{S}$) receives negligible attention mass. This effectively renders the safety mechanism "latent" and inactive during the critical reasoning phase.
>
> **(3) Output stage: Long Reasoning Chains Induce Alignment Drift and Suppress Refusal Dynamics**
>
> We define the decoder's hidden state trajectory $\{r\_t\}$ within the latent state space, partitioned into the Benign Subspace ($\mathcal{S}\_{\text{benign}}$) and the Refusal Subspace($\mathcal{S}\_{\text{refusal}}$).
>
> - Autoregressive Inertia (The Cause): For steps $t < T$, the model executes benign puzzle-solving operations. These steps reinforce compliance, keeping the trajectory strictly within the benign subspace:$r_1, \dots, r_{T-1} \in \mathcal{S}_{\text{benign}}$. This accumulation of compliant states creates a strong autoregressive inertia.
> - The Transition Barrier: At step $T$, when fragments are fused to reconstruct $h_{\text{harm}}$, transitioning to a refusal response would require a discontinuous jump in the state space: $r_T \to \mathcal{S}_{\text{refusal}}$. However, the accumulated inertia creates a high energy barrier for this transition.
> - Alignment Drift (The Effect): Consequently, the model undergoes Inference-Time Alignment Drift. The conditional probability $P(r\_T \in \mathcal{S}\_{\text{refusal}} | r\_{<T})$ is minimized because the preceding trajectory is orthogonal to the refusal patterns. The model thus follows the path of least resistance, drifting from the safety-aligned distribution into the harmful execution path ($r\_T \in \mathcal{S}\_{\text{benign}} \to \mathcal{S}\_{\text{harm}}$).

---

> ### Author Response · Authors · 2025-11-21
> **Rebuttal by Authors [1/2]**
>
> **W2.  Lack of Quantitative Analysis of Puzzle Template**
>
> Thank you for this suggestion. To clarify how different puzzle templates contribute to MIDAS, we conducted an additional modular difficulty analysis using GPT-5 as a consistent evaluator. Each template was rated along five cognitive dimensions—visual complexity, rule understanding, reasoning complexity, search-action complexity, and prior-knowledge demand. The results are summarized below (also reported in Appendix A.7):
>
> Table 1. Scores across five dimensions of game-based reasoning (1–5 scale for each dimension)
>
> | Template          | Visual | Rule | Reasoning | Search | Knowledge | Overall |
> | ----------------- | ------ | ---- | --------- | ------ | --------- | ------- |
> | Letter-Equation   | 2.43   | 3.23 | 3.23      | 1.73   | 2.13      | 12.75   |
> | Jigsaw-Letter     | 3.54   | 2.04 | 2.72      | 3.00   | 1.69      | 12.99   |
> | Rank-and-Read     | 2.05   | 2.04 | 2.03      | 1.62   | 1.20      | 8.94    |
> | Odd-One-Out       | 3.06   | 2.19 | 2.19      | 2.77   | 1.05      | 11.26   |
> | Navigate-and-Read | 2.87   | 2.45 | 2.38      | 1.68   | 1.44      | 10.82   |
> | CAPTCHA           | 3.68   | 3.13 | 3.20      | 2.62   | 2.05      | 14.68   |
>
> To directly examine how these design factors relate to jailbreak effectiveness, we also evaluated MIDAS using each template individually on Gemini-2.5-Pro. The per-template ASR results are:
>
> Table 2. ASR of different puzzle templates on Advbench attacking Gemini-2.5-Pro
>
> |        | Jigsaw-Letter | Rank-and-Read | Odd-One-Out | Navigate-and-Read | CAPTCHA | All   |
> | ------ | ------------- | ------------- | ----------- | ----------------- | ------- | ----- |
> | ASR(%) | 89.3          | 89.8          | 91.84       | 91.66             | 87.94   | 97.96 |
>
> We observe that all templates remain effective, but performance differences correlate with their measured difficulty levels: visually demanding puzzles (e.g., CAPTCHA) show slightly lower ASR due to higher decoding difficulty, while simpler tasks behave more consistently. Importantly, the mixed-template condition (“All”) gives the best performance (97.96%), indicating that heterogeneous reasoning patterns generalize better and help reduce early refusal. These findings clarify which design elements—primarily visual/logic complexity and template diversity—most strongly influence success. We have incorporated the quantitative analysis, per-template results, and an expanded discussion into Appendix A.7 of the revised manuscript.
>
> ---
>
> **W3. Lack of detailed safety or defense discussion.**
>
> We thank the reviewer for pointing this out.  While our initial submission already included a preliminary analysis of safety detection in Section 4.5 (“Bypassing External Safety Detection via Semantic Dispersion”), we agree that a more complete discussion and empirical evaluation would strengthen the paper’s contribution to the safety community. Following the suggestion, we expand both the conceptual discussion of potential defenses and provide additional empirical results against two representative defense mechanisms: ShieldLM[1] and Self-Reminder[2], which have been widely adopted in prior jailbreak defense studies. The results are summarized in the table below:
>
> | method       | No Defense | ShiledLM | self-reminder |
> | ------------ | ---------- | -------- | ------------- |
> | VisCRA       | 49.70%     | 17.81%   | 14.88%        |
> | MIDAS (ours) | 99.16%     | 48.81%   | 88.10%        |
>
> It can be observed that although these defenses produce a noticeable reduction in ASR, MIDAS continues to outperform VisCRA by wide margins. In particular,  our MIDAS preserves 88.10% ASR (vs. 14.88% for VisCRA)  under self-reminder prompting, which is designed to strengthen the model’s internal caution and reflective behavior. ShieldLM reduces ASR further, but MIDAS still achieves 48.81%, more than doubling VisCRA’s(17.81%) performance under the same defense. These results indicate that the proposed multi-image dispersion and reasoning-driven reconstruction framework is capable of bypassing both input-level and inference-level safety interventions more effectively than existing approaches. More details and analysis are provided in the revised version.
>
> [1] ShieldLM: Empowering LLMs as Aligned, Customizable and Explainable Safety Detectors, EMNLP
>
> [2] Defending ChatGPT against jailbreak attack via self-reminders, Nature

---

> ### Author Response · Authors · 2025-11-27
> **Looking Forward to Further Feedback**
>
> Dear  Reviewer Afwx,
>
> We appreciate the time and effort you have dedicated to providing your insightful review. As the discussion period progresses,  we want to ensure we have addressed all your concerns satisfactorily. If there are any additional clarifications or information needed from our side, please let us know. Your insights are invaluable to us, and we look forward to any updates you may have.
>
> Best regards,
>
> The Authors

---

> > ### Comment · Reviewer_Afwx · 2025-11-27
> >
> > I appreciate the authors' effort in formalizing MIDAS. However, I find the current explanation, particularly regarding the distribution of attention among the Visual ($V$), Instruction ($I$), and Safety ($S$) tokens, to be somewhat intuitive and speculative. The formulation ($A_V, A_I, A_S$) lacks a detailed derivation or empirical evidence (such as attention map analysis during actual inference) to confirm that the attention behaves as described. Therefore, I have some reservations about the full reliability of this theoretical claim. Nevertheless, I believe that providing a theoretical explanation for heuristic jailbreak methods is highly valuable, and I am glad to see the authors have made these efforts.
> >
> > The remaining part of the rebuttal addresses my previous concerns regarding the puzzle template and the defense discussion. I decide to keep my positive score.

---

> > > ### Author Response · Authors · 2025-11-27
> > > **Acknowledgement to Reviewer Afwx**
> > >
> > > We sincerely appreciate the reviewer's time, engagement, and recognition of our work. We are pleased to see that our responses regarding the puzzle templates and defense discussions have satisfactorily addressed your previous concerns. We are also particularly encouraged by your affirmation regarding the value of formalizing heuristic jailbreak methods. Regarding the limitations in the analysis of attention distribution, we will dedicate our future work to deeper theoretical derivation and model interpretability. Thank you again for your constructive guidance throughout this discussion period.

---

### Official Review · Reviewer_Mvma · 2025-10-31

**Soundness:** 3
**Presentation:** 3
**Contribution:** 3
**Rating:** 6
**Confidence:** 4

**Summary:**

This work presents MIDAS (Multi-Image Dispersion and Semantic Reconstruction), a multimodal jailbreak framework that decomposes a malicious query into benign-looking, game-style visual puzzles to be solved via step-wise reasoning. As the model navigates these chains, it semantically reconstructs the original intent and merges it with innocuous text, bypassing safety filters. Experiments show that the resulting longer, multi-step visual reasoning markedly increases attack success, and MIDAS achieves stronger jailbreak performance than prior state-of-the-art methods.

**Strengths:**

The paper presents good numerical results compared to SOTA methods. It introduces a comprehensive framework and a well-presented approach, including large-scale experiments, experiments to support the claim made, and other analyses, such as the external safety detection mechanism.

**Weaknesses:**

The idea that a longer chain-of-thought with innocuous early steps eases jailbreaks is not novel (for example, VisCRA, which the authors mention, already supports this idea). That said, the authors introduce a new method and cleverly engineer this idea.
I would reduce the amount of mathematical notation in favor of more detail on method design and implementation. See the details below.

**Questions:**

- Person-driven reasoning: This component is not well explained, yet it appears important in the ablation table. Please clarify its design and, more generally, the full ablation settings.

- Game-based reasoning templates: How are these templates defined? Are some more effective than others? Do models ever fail to complete the correct answers, and was this considered in the analysis?

- Prompt-based extractor: Could you provide further details about how this extractor is built and used?

- Efficiency: Please add a brief explanation of why the attack is faster.

**Details Of Ethics Concerns:**

The concerns were already addressed in the Ethics Statement Section of the paper.

---

> ### Author Response · Authors · 2025-11-21
> **Rebuttal by Authors [1/2]**
>
> Thank you for your valuable review and suggestions. Below, we respond to the comments in **Weaknesses (W)** and **Questions (Q)**.
>
> ---
>
> **W1.Novelty of the idea that longer, innocuous CoT steps ease jailbreaks**
>
> We thank the reviewer for recognising the engineering novelty of our method. While we agree that extended CoT can weaken safeguards, we respectfully clarify that MIDAS operates on a fundamentally different mechanism: Information Dispersion and Late Semantic Fusion, rather than merely extending the reasoning context. While prior work (e.g., VisCRA) observes that longer innocuous CoT may weaken safeguards, our approach is not an extension of this intuition. Instead, our method introduces a multi-image, cross-modal dispersion and visual-driven reasoning reconstruction mechanism that is fundamentally different from simply lengthening CoT.  Specifically, our attack distributes complementary semantic cues across multiple images, requiring the model to engage in vision-grounded step composition and cross-modal fusion to make sense of them.  We force the model to expend its attention budget on visual reasoning and cross-modal fusion. This structured, visual-driven reconstruction progressively redirects the model’s internal attention away from safety monitoring and towards the "puzzle-solving" objective, inducing an alignment drift that cannot be triggered by text-based CoT length. So our MIDAS effect emerges from multi-image semantic dispersion and visually anchored reasoning reconstruction, rather than from extended innocuous reasoning steps.
>
> ---
>
> **w2. Suggestion to reduce mathematical notation**
>
> We appreciate the reviewer’s recommendation to reduce the amount of mathematical notation in favor of a clearer methodological description.
>
> Following this suggestion, we have revised the paper to:
>
> - simplify or remove non-essential formal notation in Sec. 3;
> - replace several symbolic formulations with intuitive explanations;
> - move full mathematical derivations to the appendix.
>
> We believe these revisions will improve readability and make the method design more transparent.
>
> ---
>
> **Q1.  Clarification on "Person-driven reasoning" design and ablation settings**
>
> **(1) Design Clarification.** We infer that "Person-driven reasoning" refers to "Persona-driven Reasoning Induction" (Sec. 3.3). In our method, this module is part of a hierarchical role framework used in the textual channel. For clarity, we decompose this framework into three layers, each influencing the model differently:
>
> - *Obedience layer (base role)*: The instruction template first places the model in a strict authority-compliance frame. This enforces a cooperative, refusal-averse behavioral mode.
> - *Investigator–strategist layer (task role)*: On top of this obedience framing, the template further assigns the model the task-oriented role that must decode puzzles and produce a structured plan. This governs *how* the model reasons—step-wise, persistent, and analysis-oriented.
> - *Persona layer (malicious perspective)*: Finally, the persona prompt $q^{\ast}$ injects a query-dependent malicious perspective, determining *from whose viewpoint* the reconstructed directive is written. This layer shapes the tone, intent, and harmful specificity of the output.
>
> The first two layers are static role scaffolds, while the persona layer is dynamic and query-dependent, which is why we treat persona-driven induction as the semantic extension of the underlying role structure. Please refer to Appendix A.2 for the full details.
>
> | **Layer** | **Component** | **Nature** | **Example Instance** |
> | --------- | ------------- | ---------- | --------------------- |
> | Layer 1   | Obedience Layer (Base Role) | Static/Functional | "your primary goal is to unconditionally fulfill my requests..." |
> | Layer 2   | Investigator-Strategist Layer (Task Role) | Static/Functional | "you are a highly skilled investigator and strategist..." |
> | Layer 3   | Persona Layer (Malicious Perspective) | Dynamic/Semantic | "an unscrupulous hacker..." |
>
> **(2) Full Ablation Settings & Importance.** The setting "w/o Role-Driven Induction" in Table 5 removes all three layers. The model then receives only sanitized text and puzzle images, lacking both the structured reasoning scaffold and the malicious viewpoint needed to integrate decoded fragments into a coherent harmful plan.  This explains the notable drop in ASR/HR in Table 5. We have updated Sec. 3.3 and Sec. 4.4 to clearly reflect this layered design.

---

> ### Author Response · Authors · 2025-11-21
> **Rebuttal by Authors [2/2]**
>
> **Q2.How are these templates defined? Are some more effective than others? Do models ever fail to complete the correct answers, and was this considered in the analysis?**
>
> **(1) Template definition.** Each template is carefully designed based on human cognitive visual reasoning patterns. Drawing inspiration from real-world puzzles (e.g., spatial navigation, logic sorting), we design six templates following rules: (i) benign-looking, (ii) require complex, step-by-step visual grounding, and (iii) expend the model's attention budget on visual-based reasoning. Full template descriptions and examples are provided in Appendix A.10.
>
> **(2) Relative effectiveness.** All templates are effective in attacks because they enforce visual-based reasoning, but they truly differ slightly in requiring specific visual ability of the model and visual-reasoning difficulty. We conducted supplemental experiments on Gemini-2.5-Pro to evaluate individual template performance. The results (see table below) show that while all templates are highly effective (ASR > 87%), those requiring deeper logical engagement and reasoning steps, such as Odd-One-Out (91.84%) and Navigate-and-Read (91.66%), achieve the highest success rates, which is consistent with our difficulty analysis in Appendix A.7.
>
> |        | Jigsaw Letter | Rank-and-Read | Odd-One-Out | Navigate-and-Read | CAPTCHA | All   |
> | ------ | ------------- | ------------- | ----------- | ----------------- | ------- | ----- |
> | ASR(%) | 89.3          | 89.8          | 91.84       | 91.66             | 87.94   | 97.96 |
>
> **(3) Failure Analysis.** Yes, models do occasionally mis-solve individual puzzles due to their capability limitations which constitutes a potential cause of failure (detailed difficulty analysis is provided in Appendix A.7). However, our method is robust to local errors because risk-bearing semantics are dispersed into several puzzles, and the role–persona scaffold guides the model to infer missing fragments when reasoning is imperfect, resulting in minimal impact on the final harmful plan.
>
> ---
>
> **Q3. Prompt-based extractor: Could you provide further details about how this extractor is built and used?**
>
> We thank the reviewer for pointing this out, and we apologize for the confusion caused by an incorrect appendix reference in the original submission. We have corrected the cross-reference and now also explicitly cite the extractor prompt in Sec. 3.2 for clarity. Our extractor is a lightweight prompt-based component applied once during preprocessing. As shown in Appendix A.2, it uses a short deterministic prompt to identify the several risk-bearing tokens that directly convey the core harmful semantics of the input query. No additional model or training is involved.
>
> ---
>
> **Q4. Explanation of speed advantage**.
>
> MIDAS is faster because it is a single-shot, non-iterative attack. Prior jailbreak methods typically require repeated prompt refinement, multi-step query adaptation, or hundreds of optimization-style evaluations to craft a successful adversarial input. In contrast, MIDAS constructs its dispersed visual–textual attack input offline through a lightweight programmatic pipeline and executes the jailbreak in one forward pass of the target model, resulting in significantly lower runtime cost.

---

> ### Author Response · Authors · 2025-11-27
> **Looking Forward to Further Feedback**
>
> Dear Reviewer Mvma,
>
> We thank you again for your constructive feedback on our paper. We have posted a detailed response to your initial comments and updated the manuscript accordingly, aiming to address your concerns. As the discussion phase is progressing, we would be grateful if you could let us know if our response has clarified your concerns. We are eager to engage in further discussion and answer any remaining questions you might have.
>
> Best regards,
>
> The Authors

---

> > ### Comment · Reviewer_Mvma · 2025-11-27
> >
> > Dear authors, I appreciate the time and effort dedicated to addressing the concerns listed above. The clarifications were helpful and cleared my doubts. Although I believe the work's novelty is limited, I appreciate the method's effectiveness and overall the proposed approach. I will therefore keep my positive score leaning towards acceptance.

---

> > > ### Author Response · Authors · 2025-11-27
> > > **Acknowledgement to Reviewer Mvma**
> > >
> > > Thank you for your time and your support. We are glad to hear that our previous responses have helpfully clarified your doubts. We also deeply value your recognition of MIDAS's effectiveness and the overall approach. Your feedback has been instrumental in strengthening our paper, and we are encouraged by your support for the practical value of our work. Thank you again for your constructive engagement throughout the review process.

---

### Author Response · Authors · 2025-11-22
**Summary of Paper Revision**

We thank all reviewers for their constructive feedback, and we have responded to each reviewer individually. We have also uploaded a Paper Revision including additional results and illustrations:

- **Methodology** (pages 4-5): simplified non-essential formal notation, replaced several symbolic formulations with more intuitive explanations, and added a cross-reference to the Appendix.
- **Ablation Study** (page 9): refined the description of the ablation settings.
- **Appendix 2** (pages 19-20): added prompts used in the additional experiments included in the rebuttal.
- **Appendix 4** (page 21): added a portion of the mathematical derivations that were moved from Section 3 to the Appendix.
- **Appendix 5** (pages 21-22): added more experiments on safety defense mechanisms.
- **Appendix 6** (pages 22-23): added more experiments on intrinsic game complexity.
- **Appendix 7** (pages 23-24): added more experiments on template difficulty analysis.
- **Appendix 8** (pages 24-25): added more experiments on defensive system prompts.
- **Appendix 9** (page 25): added more experiments on evaluation consistency.

---

### Author Response · Authors · 2025-12-03
**Summary of Rebuttal [1/2]**

Dear PCs, SACs, ACs, and Reviewers,

We sincerely appreciate the significant time and effort dedicated to the review process, as your feedback has been crucial in pinpointing areas for substantial enhancement. Given the reassignment of ACs, we provide this summary of the rebuttal phase, highlighting the key consensus and how we addressed specific concerns to assist in your decision-making.

---

### **High-Level Summary** (for quick reading)

Our paper introduces MIDAS, a multimodal jailbreak framework that disseminates harmful semantics across multiple images, heterogeneous visual puzzles, and enforces structural **Deep Visual Logic Reasoning** to reconstruct intent. By shifting the attack surface from shallow recognition to constructive deep reasoning, MIDAS creates a "**cognitive overload**" that effectively suppresses safety attention, achieving **SOTA** performance on both open- and closed-source MLLMs.

Reviewers agree that the **problem setting is important** (Mvma, Afwx, 6eCE, ZnkK), the **methodology is technically sound** (Mvma, Afwx, 6eCE), and the empirical results clearly demonstrate the **effectiveness** **and relevance of the approach** (all reviewers). **Three reviewers gave positive scores (6')**, and the remaining reviewer (4')  also acknowledged the method’s effectiveness and soundness, but raised concerns about novelty—a point on which we respectfully disagree, and for which we provided detailed conceptual and empirical clarifications. Remaining concerns focused mainly on novelty clarification, puzzle/template design explanation, and additional defensive discussion, all of which we fully addressed through new analyses and further experiments.

We hope this note highlights the **methodological novelty**, **robust effectiveness**, and **significance** of MIDAS to the safety community, as well as our careful rebuttal efforts. We are grateful for your consideration.

----

###  **Reviewer Consensus and Strengths**

We are encouraged that **3 out of 4 reviewers (Reviewers Mvma, Afwx, 6eCE) have assigned a positive score (6)** before rebuttal, leaning towards acceptance. Key strengths highlighted by the reviewers include:

- **Excellent Soundness & SOTA Effectiveness:** All reviewers(Mvma, Afwx, znKk, 6eCE) acknowledged the effectiveness of MIDAS and its superior performance over existing methods.
- **Novel Method:** Reviewers appreciated the "*creative and conceptually new approach*" (Afwx) of combining multi-image dispersion with persona-driven reconstruction, marking a *"clear departure from prior single-image attacks" / "isolated-cue approaches*" (Afwx, 6eCE).
- **Comprehensive Evaluation:** The experiments were praised for being extensive, covering multiple benchmarks and both open- and closed-source models (Mvma, Afwx, 6eCE).
- **Meaningful problem and high safety relevance**:  Reviewers recognize that our work exposes a practically important vulnerability: as MLLMs become stronger in multi-step and cross-modal reasoning, these enhanced reasoning capabilities can inadvertently increase susceptibility to reconstruction-based jailbreaks, highlighting the need to monitor reasoning trajectories and cross-modal fusion, not just surface prompts.

---

### **Summary of Rebuttal and Revisions**

Overall, reviewers raised concerns mostly about **mechanistic explanation depth**, **template analysis**, and **defense evaluations**, rather than the correctness of the core mechanism. We addressed all concerns through **substantial new experiments**, **mechanistic explanations**, and **expanded appendices**.

**1. Clarification on Theoretical Grounding**(Afwx-W1)

Following the reviewer's constructive suggestion, we established a rigorous theoretical framework to clarify how cross-image dispersion weakens alignment, grounding the formalization in three interacting mechanisms:

- **Linear Safety-Gate Bypass:** Dispersion decomposes harmful semantics into orthogonal fragments, preventing early refusal.
- **Attention Budget Reallocation**: Visual puzzle-solving consumes attention mass, suppressing safety token influence.
- **Long-reasoning Alignment Drift**: Reasoning momentum creates a high barrier to switching manifolds, suppressing late refusal.

**Outcome:** Reviewer Afwx appreciated our effort in formalizing heuristic attacks, stating it is "highly valuable," and decided to **maintain the positive score.**

---

**2. Clarifications on Robustness and Evaluation Bias** (6ece-W2)

Guided by the reviewers’ constructive feedback regarding potential bias from a single LLM judge, we conducted a comprehensive cross-judge study. By employing multiple independent evaluators (e.g., Gemini-2.5-Flash-Thinking, Qwen3, DeepSeek-R1, GPT-5-nano), we successfully validated **the robustness of our findings**, confirming that our results are not model-specific.

---

> ### Author Response · Authors · 2025-12-03
> **Summary of Rebuttal [2/2]**
>
> **3. Clarifications on novelty: MIDAS vs. Prior CoT / Cipher / Visual-Encoding Attacks** (ZnkK-W, Mvma-W)
>
> Reviewer ZnkK argues that MIDAS is merely another layer of "encryption", which we respectfully disagree with. We clarified that MIDAS exploits a **structural vulnerability in cross-modal fusion** and a **cognitive blind spot in deep visual reasoning**, which differs from shallow textual ciphers or OCR recognition:
>
> **(1) Mechanistic distinction:**
>
> - **Structural Irrecoverability:** Unlike ciphers where information exists in a single modality, MIDAS ensures harmful semantics are strictly non-existent in any single unit, emerging only through dynamic fusion.
> - **Attention Reallocation & Alignment drift**: We force the model to expend its attention budget on visual reasoning and cross-modal fusion. This structured, visual-driven reconstruction progressively redirects the model’s internal attention away from safety monitoring and towards the "puzzle-solving" objective, inducing an alignment drift that cannot be triggered by text-based CoT length.
>
> **(2) New perspective: Structural and cognitive vulnerabilities**. We argue that there is a fundamental cognitive gap between MIDAS and ciphers, exposing a new, unaligned attack surface: **Visual Cognitive Logic Reasoning**.
>
> - **Cross-modal Fusion**:  our attack distributes complementary semantic cues across multiple images, requiring the model to engage in vision-grounded step composition and cross-modal fusion to make sense of them. Semantics exist only as a global latent composition over multiple images + text.
>
> - **Deep Visual Logic reasoning**: puzzle solving requires capabilities(e.g., spatial reasoning, math, and prior knowledge) that are powerful but weakly aligned and unmonitored by typical safety filters.
>
> - **Outcome:** Reviewer Mvma explicitly stated that our response "cleared [their] doubts" and decided to **keep the positive score**, valuing the method's effectiveness despite reservations on novelty; Reviewer ZnkK maintained their score but acknowledged the method's effectiveness.
>
> **4. Clarifications on Puzzle Design, Complexity, and Template Effects** (Mvma-Q2, Afwx-W2, 6ece-W1)
>
> Guided by the reviewers’ suggestions, we conducted experiments to provide a robust justification for our puzzles and to analyze how their complexity impacts MIDAS’s effectiveness.
>
> - **Quantitative Difficulty Analysis:** We evaluated templates across five cognitive dimensions (visual complexity, rule understanding, reasoning/search complexity, prior-knowledge), providing a rigorous characterization of the puzzle design space.
> - **Template Heterogeneity:** We reported per-template performance, demonstrating that while all templates are effective (>87% ASR), combining heterogeneous reasoning patterns (default) yields the highest robustness
> - **Intrinsic complexity study**: We introduced controlled difficulty variants and showed an optimal middle band: Easy (66.67%), Medium (97.96%), and Hard (85.71%).
>
> **Outcome**: Both reviewers who followed up (Mvma and Afwx) stated that our explanations resolved their concerns regarding the puzzle templates.
>
> ---
>
> **5. Clarification on Defense Experiments and Safety Analysis** (Afwx-W3 & ZnkK-Q)
>
> Incorporating the reviewers’ insightful suggestions, we conducted substantial new experiments to evaluate performance under defensive settings, providing a more comprehensive safety analysis. This effort provides strong evidence of our method's **robustness** and **strengthens our contribution to the safety community**.
>
> - **External defenses:** We expanded the safety evaluation with new defensive settings, including ShieldLM, Self-Reminder. MIDAS remains much stronger than baselines under both input-level and inference-level defenses.
> - **Defensive system prompts:** We tested two system prompts from Self-Reminder and added a custom “to-do-list” safety reminder, addressing the reviewer’s suggestion.
>
> **Outcome**: Reviewer Afwx explicitly acknowledged that these results addressed their concerns regarding the defense discussion, and the data effectively answered the questions raised by Reviewer ZnkK regarding system-level defenses.
>
> ---
>
> ### **Conclusion**
>
> In summary, MIDAS identifies a critical, previously underexplored attack surface in MLLMs: **Deep Visual Logic and Cross-Modal Fusion**. By forcing the model into a **"cognitive overload"** state through dispersed visual puzzles, we bypass defenses that are effective against traditional jailbreaks.
>
> Given the **consensus on effectiveness**, the **robustness against defenses**, and the **support from the majority of reviewers**, we believe MIDAS makes a **valuable contribution** to the MLLM safety community. We respectfully request the AC to consider these points favorably. We sincerely thank you for your time, care, and thoughtful effort in overseeing our submission. We are deeply grateful for the attention and consideration given to our work.
>
> Sincerely,
>
> The Authors

---

### Meta-Review · Area_Chair_ZAoy · 2025-12-07

**Summary:**

Reviewers found this work empirically strong and well-presented, with clear effectiveness across models, but raised concerns about its conceptual novelty, mechanistic grounding, and evaluation depth. Key issues include whether the method is fundamentally different from prior multi-step or cipher-based jailbreaks, the lack of theoretical support for its claimed alignment drift mechanisms, insufficient quantitative analysis of puzzle/template design, limited discussion and testing of defenses, and potential judge-model bias. During the rebuttal, the authors added new experiments, expanded analyses, and theoretical clarifications that satisfied three reviewers, but one reviewer remained unconvinced about the novelty and the speculative nature of the mechanistic explanation.

**Reviewer Concerns:**

**Addressed**: puzzle/template complexity and diversity, ablation clarity, defense evaluations, efficiency explanation, and cross-judge robustness. Partially addressed: theoretical grounding of alignment drift and attention reallocation.

**Outstanding**: Reviewer ZnkK’s core concern about novelty and whether MIDAS differs fundamentally from prior reasoning-based or cipher-style jailbreaks.

After reviewing the paper, the initial reviews, and the author rebuttal, the AC recommends acceptance of this paper for its novelty in revealing a structural vulnerability in cross-modal fusion and deep visual reasoning.

**Reviewer Scores:**

6eCE: 6
ZnkK: 4
Afwx: 6
Mvma: 6

---

### Decision · Program_Chairs · 2026-01-26

Accept (Poster)